# Chemical and climatic drivers of radiative forcing due to changes in stratospheric and tropospheric ozone over the 21[st] century

A. Banerjee[1], A. C. Maycock[2], and J. A. Pyle[3,4]

[1]Department of Applied Physics and Applied Mathematics, Columbia University, New York, NY, USA.
[2]School of Earth and Environment, University of Leeds, Leeds, UK.
[3]Department of Chemistry, University of Cambridge, Cambridge, UK.
[4]National Centre for Atmospheric Science - Climate, UK.

*Correspondence to*: Antara Banerjee (ab4283@columbia.edu)

**Abstract.** The ozone radiative forcings (RFs) resulting from projected changes in climate, ozone-depleting substances (ODSs), non-methane ozone precursor emissions and methane between the years 2000 and 2100 are calculated using simulations from the UM-UKCA chemistry-climate model. Projected measures to improve air-quality through reductions in non-methane tropospheric ozone precursor emissions present a co-benefit for climate, with a net global mean ozone RF of -0.09 W m$^{-2}$. This is opposed by a positive ozone RF of 0.05 W m$^{-2}$ due to future decreases in ODSs, which is driven by an increase in tropospheric ozone through stratosphere-to-troposphere transport of air containing higher ozone amounts. An increase in methane abundance by more than a factor of two (as projected by the RCP8.5 scenario) is found to drive an ozone RF of 0.18 W m$^{-2}$, which would greatly outweigh the climate benefits of tropospheric non-methane ozone precursor reductions. Asmall fraction (~15%) of the ozone RF due to the projected increase in methane results from increases in stratospheric ozone. The sign of the ozone RF due to future changes in climate (including the radiative effects of greenhouse gas concentrations, sea surface temperatures and sea ice changes) is shown to be dependent on the greenhouse gas emissions pathway, with a positive RF (0.05 W m$^{-2}$) for RCP4.5 and a negative RF (-0.07 W m$^{-2}$) for the RCP8.5 scenario. This dependence arises from differences in the contribution to RF from stratospheric ozone changes. Considering the increases in tropopause height under climate change causes only small differences ($\leq$|0.02| W m$^{-2}$) for the stratospheric, tropospheric and whole-atmosphere RFs.

## 1 Introduction

Ozone is a so-called secondary pollutant, being primarily formed by chemical processes within the atmosphere rather than being emitted directly at the surface. Emissions into the atmosphere of well-mixed greenhouse gases (WMGHGs - e.g. $CO_2$, $CH_4$, $N_2O$, CFCs), ozone-depleting substances (ODSs - CFCs and other halogenated species controlled by the Montreal Protocol) and tropospheric ozone precursors (e.g. $CH_4$, $NO_x$, CO) all modify concentrations of ozone. Thus, the total radiative forcing (RF) due to the emission of a specific gas into the atmosphere may include an indirect component through ozone, in addition to any radiative forcing associated with the gas itself (e.g. Myhre et al., 2013).

Emissions-based estimates of pre-industrial to near present-day (1750-2011) ozone RFs (with 5-95% confidence ranges) are -0.15 (-0.30 to 0.00) W m$^{-2}$ due to ODSs and 0.50 (0.30 to 0.70) W m$^{-2}$ due to ozone precursors (Myhre et al., 2013). This can be compared to a WMGHG forcing of 2.83 (2.54 to 3.12) W m$^{-2}$ over the same period (Myhre et al., 2013). The emission-based estimates of historical ozone RF in Myhre et al. (2013) include the effects of changes in both stratospheric and tropospheric ozone. The historical ozone RF due to ODS emissions has been largely due to changes in stratospheric ozone abundance. Correspondingly, the ozone RF from ozone precursors has been largely due to changes in its tropospheric abundance. However, the emissions of such species that affect ozone abundances can also exert a significant influence on ozone away from their region of primary impact, for example through effects on stratosphere-to-troposphere exchange (STE) of ozone (Shindell et al., 2013a; Søvde et al., 2011). The tropospheric ozone RF due to the effects of past changes in ODSs is estimated to be about one third to one quarter of the stratospheric RF. Similarly, for past changes in ozone precursors, the stratospheric ozone RF is estimated to be ~15-20% of the tropospheric ozone RF. However, the relative contributions to RF of stratospheric and tropospheric ozone under future ozone recovery, owing to the phase out of ODSs, remain to be quantified. It also remains to be determined which of the ozone precursors - $CH_4$, $NO_x$, CO or non-methane volatile organic compounds (NMVOCs) - affect stratospheric ozone RF, and how this will evolve in the future.

The Representative Concentration Pathway (RCP) scenarios for future anthropogenic emissions adopted in IPCC (2013) project reductions in emissions of air pollutants including non-methane ozone precursors (van Vuuren et al., 2011). Any reductions in tropospheric ozone abundances that occur as a result represent a co-benefit to climate (e.g. Fiore et al., 2008). However, there are added complications of further climate impacts through changes in concentrations of nitrate aerosol and the hydroxyl (OH) radical (Myhre et al., 2013); only the latter effect is explored in this study. Changes in OH concentration perturb the $CH_4$ lifetime and its steady state abundance (e.g. Fuglestvedt et al., 1999). Steady state ozone abundances are also affected by changes in $CH_4$ lifetime since $CH_4$ is a major tropospheric ozone precursor (Crutzen, 1973). Accounting for adjustments through changes in the $CH_4$ lifetime can lead to a net climate penalty under reductions of $NO_x$ emissions if the direct RF due to resulting changes in $CH_4$ is included along with the associated RF from changes in ozone (Naik et al., 2005). In contrast, $CH_4$ adjustments can result in a greater climate benefit under CO and NMVOC emission reductions (e.g. West et al., 2007; Stevenson et al., 2013). The RCP8.5 scenario assumes a particularly large increase in $CH_4$ by 2100 (van Vuuren et al., 2011), the effect of which swamps the tropospheric ozone RFs of $NO_x$, CO and NMVOCs (Myhre et al., 2013). Given their distinct projected trajectories, this study seeks to isolate the ozone RF of non-methane ozone precursors from that of $CH_4$ in the RCP8.5 scenario.

Most studies that have calculated the ozone RF from changes in future climate (defined here as the radiative effects of WMGHGs, including feedbacks through surface temperature and sea ice changes) have explored only a single WMGHG emissions scenario. For example, a recent chemistry-climate model (CCM) inter-comparison study suggests a tropospheric ozone RF of $-0.033 \pm 0.042$ W m$^{-2}$ (multi-model mean $\pm$ 1σ) due to climate change up to 2100 under the RCP8.5 scenario, which is a negligible change from the forcing in the year 2000 of $-0.024 \pm 0.027$ W m$^{-2}$ (both relative to 1850) (Stevenson et al., 2013). Portmann and Solomon (2007) used the SRES A2 scenario (IPCC, 2007) (which lies between RCP6.0 and

RCP8.5 in terms of $CO_2$ concentration in the latter half of the 21[st] century) and calculated a stratospheric ozone RF of -0.08 W m$^{-2}$ due to the $CO_2$ change between 2000 and 2100. However, ozone RFs are highly sensitive to the vertical profile of ozone changes (Lacis et al., 1990), which show a strong dependency on the greenhouse gas emissions scenario, particularly in the tropics (Banerjee et al., 2016; Eyring et al., 2013). The RF due to future changes in ozone might therefore be expected to be sensitive to the emissions scenario and this warrants investigation.

The aim of this study is to quantify the indirect RFs resulting from changes in stratospheric and tropospheric ozone abundances between year 2000 and 2100 using simulations from a state-of-the-art CCM and offline radiative transfer calculations. The ozone changes are obtained from perturbations made individually to the following drivers (i) the physical climate (i.e. the radiative effects of WMGHGs), following the RCP4.5 and RCP8.5 scenarios, (ii) ODSs, (iii) non-methane ozone precursor emissions, and (iv) $CH_4$. The chemical impacts of $N_2O$ are not investigated in this study although its radiative effects on climate is implicitly contained in (i). However, we note that changing concentrations of $N_2O$ within the RCP scenarios is also expected to impact on ozone, and hence be associated with an indirect RF in the stratosphere (Butler et al., 2016; Fleming et al., 2011; Portmann and Solomon, 2007; Revell et al., 2012). Most of the model studies addressing future indirect RFs due to ozone conducted thus far have contained comprehensive chemistry in either the stratosphere or in the troposphere, but not both (Portmann and Solomon, 2007; Stevenson et al., 2013), which partly motivates this study. Here, the strength lies in the whole-atmosphere chemical scheme employed in the CCM, enabling a more complete quantification of the contributions of stratospheric and tropospheric ozone to future RF. In addition, unlike most previous studies which assume a single future WMGHG forcing scenario (e.g. Portmann and Solomon, 2007; Stevenson et al., 2013), this study quantifies the dependence of the ozone RF on two scenarios of climate change (RCP4.5 and RCP8.5).

## 2 Methods

### 2.1 Calculations of ozone response

Changes in atmospheric ozone abundances (year 2100 vs. 2000) due to future perturbations in radiative and chemical drivers have been calculated using the UK Met Office's Unified Model containing the United Kingdom Chemistry and Aerosols sub-model (UM-UKCA). The model is a stratosphere-resolving (model lid ~84 km) CCM that comprehensively describes both stratospheric and tropospheric chemistry (Morgenstern et al., 2009; O'Connor et al., 2014), with interactive ozone and water vapour. Further details of the model are provided in Banerjee et al. (2014, 2016).

Data from six time-slice experiments with fixed seasonally-varying boundary conditions are used in this study and summarized in Table 1. All but the ΔCH4 experiment are described in detail by Banerjee et al. (2016). The control experiment (Base) represents the state of the atmosphere at year 2000. The remaining five experiments perturb selected boundary conditions to year 2100 levels. Owing to computational limitations, we have not explored all possible RCP scenarios for these perturbations but rather choose a subset that is commonly explored within the literature. Experiments ΔCC4.5 and ΔCC8.5 perturb the *climate* state (i.e. including atmospheric radiative effects of WMGHGs, plus changes in sea

surface temperatures (SSTs) and sea ice) according to the medium-low (RCP4.5) and high (RCP8.5) future emissions scenarios, respectively, without changing any *chemical* boundary conditions. Here, the WMGHGs considered are $CO_2$, $CH_4$, $N_2O$, CFCs, HCFCs and HFCs. In contrast, experiments $\Delta ODS$, $\Delta O3pre$ and $\Delta CH4$ leave *climate* boundary conditions unperturbed at year 2000 conditions, but instead perturb *chemical* boundary conditions i.e. surface concentrations of ODSs, emissions of non-methane ozone precursors (from anthropogenic and biomass burning sources defined as in Lamarque et al. (2010)) and the surface concentration of $CH_4$, respectively. In this way, we distinguish the chemical and transport effects on ozone resulting from changes in the physical climate state from the chemical effects on ozone due to changes in abundance of reactive gases. All RCP scenarios project a common reduction in ODS and non-methane ozone precursor emissions, so we arbitrarily follow the RCP4.5 scenario in the $\Delta ODS$ and $\Delta O3pre$ experiments. In the $CH_4$ experiment, an increase in the $CH_4$ surface concentration by more than a factor of two (from 1.75 to 3.75 ppmv) is imposed according to the RCP8.5 scenario to explore the impact of a very large increase in $CH_4$. The initial atmospheric concentrations of ODSs and $CH_4$ were also perturbed by the same factor in $\Delta ODS$ and $\Delta CH4$, respectively, in order to reduce spin up time. In all simulations, including $\Delta O3pre$, emissions from natural sources (e.g. isoprene emissions) are non-interactive and are held fixed at year 2000 levels. In the $\Delta ODS$ run, by design, the direct radiative effect of ODSs and associated changes in physical climate state (WMO, 2014) are not captured since their concentrations are held fixed at year-2000 values within the radiation scheme. Similarly, the radiative effect of $CH_4$ on climate is not captured by design in the $\Delta CH4$ run.

There are some forcings and interactions that we do not consider in this study. Firstly, our focus lies on estimating the future ozone RF from emitted gases. We do not simulate any associated aerosol forcing, with aerosol precursor emissions and their oxidant fields being held fixed in all simulations (following the scheme of Bellouin et al. (2011)). Secondly, the 'snapshot' experiments of this study do not consider various transient interactions. For example, the background conditions of $NO_x$ and ODSs affect $CH_4$ concentrations, but this coupling is not considered when perturbing $NO_x$, ODSs and $CH_4$ individually in the $\Delta O3pre$, $\Delta ODS$ and $\Delta CH4$ experiments (potential consequences for the $CH_4$-induced ozone RF are, however, discussed in Sect. 3.4).

Each experiment was spun up for 10 years and integrated for a further 10 years. It was confirmed that this spin up period was long enough for stratospheric concentrations of perturbed gases to reach steady state. Using averages of the last 10 years, the monthly mean ozone field for each experiment is interpolated onto the Base pressure levels. The differences in ozone between Base and each perturbation experiment are then used as input to the radiative calculations.

| Experiment | Boundary conditions |
| --- | --- |
| Base | Year 2000 |
| $\Delta CC4.5$[a] | Year 2100 RCP4.5 WMGHGs in the radiation scheme only; perturbed SSTs and sea ice |
| $\Delta CC8.5$[a] | Year 2100 RCP8.5 WMGHGs in the radiation scheme only; perturbed SSTs and sea ice |
| $\Delta ODS$[b] | Year 2100 RCP4.5 ODSs in the chemistry scheme only |

| ΔO3pre [c] | Year 2100 RCP4.5 Anthropogenic and biomass burning emissions of $NO_x$, CO and NMVOCs |
|---|---|
| ΔCH4 [a] | Year 2100 RCP8.5 $CH_4$ in the chemistry scheme only |

Table 1 - List of model simulations and applied boundary conditions.

[a] WMGHGs considers the gases $CO_2$, $CH_4$, $N_2O$, CFCs, HCFCs and HFCs.

[b] ΔODS includes a total chlorine and bromine reduction at the surface of 2.3 ppbv (67 %) and 9.7 pptv (45 %), respectively.

[c] ΔO3pre includes average global and annual emission changes of NO (-51 %), CO (-51 %), HCHO (-26 %), C2H6 (-49 %), C3H8 (-40 %), CH3COCH3 (-2 %), and CH3CHO (-28 %).

[d] ΔCH4 includes an increase in the surface concentration of $CH_4$ from 1.75 to 3.75 ppmv.

## 2.2 Radiative forcing calculations

The differences in ozone abundances between year 2000 and 2100 calculated from the UM-UKCA experiments described in Section 2.1 are input to the Edwards and Slingo (1996) offline radiative transfer model (RTM) to diagnose the associated all-sky ozone RF. The model includes 9 long-wave (LW) and 6 short-wave (SW) spectral bands[1], with updates to use the correlated-k method (Cusack et al., 1999), and is the same scheme employed in the UM-UKCA model.

We calculate stratosphere-adjusted RFs using the fixed dynamical heating (FDH) method as described by Maycock et al. (2011). The calculations use monthly and zonally averaged climatologies of temperature, water vapor, ozone, WMGHGs, cloud properties, and surface albedo from the UM-UKCA Base experiment. The monthly mean year 2100 changes in ozone from each experiment are added to this background climatology, and stratospheric temperatures are adjusted using an iterative method to re-establish radiative equilibrium under the assumption that the local dynamical contribution to the heating rates does not change (IPCC, 2007). Surface and tropospheric conditions remain fixed. The RF is then diagnosed as the change in net radiative flux (downward = positive) at the tropopause. The stratospheric temperature adjustment strongly affects the calculated LW (and hence total) RF for stratospheric ozone changes, with the adjustment being largest where the SW-driven temperature changes are largest (Forster and Shine, 1997).

The lapse-rate tropopause (WMO, 1957) from the Base experiment is used for the stratospheric-adjustment and also to perform calculations to separate the RFs due to changes in tropospheric and stratospheric ozone abundances alone. In the climate change experiments, ΔCC4.5 and ΔCC8.5, the tropopause rises; the ramifications for employing a climate-consistent tropopause height for the ozone RF will be shown to be small (see Sect. 3.1). While the lapse rate tropopause is a standard measure for computing RF values, other tropopause definitions exist, including the level at which ozone equals 150 ppbv (Prather and Ehhalt, 2001). For the Base run, the climatological ozone tropopause lies very close to the thermal tropopause; for example, the tropospheric ozone burdens differ by only 2% between the two definitions. Furthermore, Stevenson et al. (2013) find less than 10% differences in the tropospheric ozone RF between 1850-2000 diagnosed in the ACCMIP models using these two tropopause definitions. Thus, for simplicity we adopt the standard lapse rate tropopause definition in this study.

---

[1] The names of the spectral files used in the RTM are for LW: spec3a_lw_hadgem1_wz_spec and for SW: spec3a_sw_hgem1_ln6e_mean_spec.

Recent studies have quantified the so-called effective radiative forcing (ERF), which accounts for rapid tropospheric adjustments (e.g. in cloud properties) resulting from the introduction of a forcing agent, in addition to the standard stratospheric temperature adjustment. A common way to calculate ERFs is to perform fixed SST global model experiments. As such, estimates of ERF are subject to statistical uncertainties arising from internal atmospheric and climate variability. Forster et al. (2016) showed that the 5-95% confidence intervals on an ERF estimated from a global climate model is around 0.1 W m$^{-2}$ for a 10 year fixed SST integration. Since the UM-UKCA experiments performed in this study are 10 years long, this would mean that the uncertainties in the estimated ERFs would, in many cases, be larger than the signal being detected. Furthermore, the differences between RF and ERF for ozone have been found to be small in previous studies (Hansen et al., 2005; Shindell et al., 2013b) and so RF is still widely adopted to assess the climate forcing from ozone (Myhre et al., 2013). For these reasons, we utilize the standard stratosphere-adjusted methodology to diagnose ozone RFs.

The radiative effects due to changes in ozone can be considered as a climate forcing mechanism (i.e. they impart a RF on climate) (Myhre et al., 2013), although in the case of the impact of changes in greenhouse gases some part of the effect may be considered as a climate *feedback* mechanism (e.g. Nowack et al., 2014). However, this distinction is not central to this study, since the UM-UKCA simulations use prescribed SSTs and sea ice and thus we wish only to quantify the net radiative effect of simulated future changes in ozone resulting from different drivers (see e.g. Stevenson et al., 2013). For simplicity, we refer to the radiative impact of simulated changes in ozone as an RF throughout the manuscript.

## 3 Results

Figure 1 shows the annual mean, global mean whole-atmosphere ozone RF (grey bars) for each perturbation experiment (see Table 1), as well as the separate contributions from changes in stratospheric (orange bars) and tropospheric (magenta bars) ozone. Figure 2 further separates the total stratospheric and tropospheric RFs into their LW (red bars) and SW (blue bars) components. Figure 3 shows the vertical profile of changes in annual mean ozone (DU km$^{-1}$) averaged over 6 latitude bands for each perturbation experiment relative to the Base run. Numerical values for each of the ozone RF components are given in Table 2. We also report the normalised radiative forcing (NRF) per unit of tropospheric ozone change (in units of W m$^{-2}$ DU$^{-1}$). This is a common measure of the tropospheric ozone RF and is estimated to be 0.042 W m$^{-2}$ DU$^{-1}$ (Myhre et al., 2013). However, we will show a wide range of NRFs between the perturbations of this study and will thus argue that it is unsuitable to arbitrarily scale NRFs across perturbations. Rather the NRF is useful in comparing the climate impacts of different perturbations through tropospheric ozone.

Figure 1 shows that, in all cases, the whole-atmosphere ozone RFs are small ($\leq$|0.2| W m$^{-2}$) compared to the combined forcing of WMGHGs between 2000 and 2100 (roughly 2 and 6 W m$^{-2}$ for RCP4.5 and RCP8.5, respectively, as shown by Fig. 10 in van Vuuren et al. (2011)). As will be discussed, some of these small whole-atmosphere RFs reflect cancellations between stratospheric and tropospheric contributions. Notably, these separate contributions are additive and equal the whole-atmosphere RFs (Table 2). The ozone distributions and the associated global mean ozone RFs for each

perturbation experiment are now discussed in Sections 3.1-3.4. The NRFs for tropospheric ozone are discussed in Section 3.5. Section 4 will discuss the latitudinal contributions to the global mean RF and seasonal variations.

| | Whole-atmosphere | | | Troposphere | | | Stratosphere | | |
|---|---|---|---|---|---|---|---|---|---|
| | LW | SW | Total | LW | SW | Total | LW | SW | Total |
| $\Delta$CC4.5 | 0.10 (0.10) | -0.05 (-0.05) | 0.05 (0.05) | 0.09 | 0.01 | 0.10 *0.040 W m$^{-2}$ DU$^{-1}$* | 0.01 | -0.06 | -0.04 |
| $\Delta$CC8.5 | -0.02 (-0.03) | -0.05 (-0.05) | -0.07 (-0.07) | 0.07 | 0.01 | 0.07 *0.069 W m$^{-2}$ DU$^{-1}$* | -0.09 | -0.05 | -0.15 |
| $\Delta$ODS | 0.39 (0.39) | -0.34 (-0.34) | 0.05 (0.05) | 0.06 | 0.01 | 0.06 *0.035 W m$^{-2}$ DU$^{-1}$* | 0.34 | -0.35 | -0.01 |
| $\Delta$O3pre | -0.08 (-0.08) | -0.01 (-0.01) | -0.09 (-0.09) | -0.09 | -0.01 | -0.10 *0.035 W m$^{-2}$ DU$^{-1}$* | 0.01 | 0.00 | 0.01 |
| $\Delta$CH4 | 0.27 (0.27) | -0.09 (-0.07) | 0.18 (0.19) | 0.14 | 0.02 | 0.15 *0.036 W m$^{-2}$ DU$^{-1}$* | 0.13 | -0.10 | 0.03 |
| $\Delta$CC8.5(f LNO$_x$) | -0.33 (-0.32) | -0.04 (-0.04) | -0.37 (-0.37) | -0.15 | -0.02 | -0.17 *0.045 W m$^{-2}$ DU$^{-1}$* | -0.17 | -0.02 | -0.19 |
| $\Delta$CC4.5(t rophgt) | 0.12 (0.12) | -0.06 (-0.06) | 0.06 (0.06) | 0.09 | 0.01 | 0.10 | 0.03 | -0.07 | -0.04 |
| $\Delta$CC8.5(t rophgt) | 0.00 (0.00) | -0.07 (-0.07) | -0.07 (-0.07) | 0.06 | 0.01 | 0.06 | -0.06 | -0.08 | -0.13 |

**Table 2. Global and annual mean ozone RFs [W m$^{-2}$] for the whole-atmosphere, troposphere and stratosphere in the different perturbation experiments. Total (LW+SW) RFs, as well as the separate LW and SW contributions, are shown. Bracketed values show the sum of the tropospheric and stratospheric values for comparison with the whole-atmosphere values. For the total tropospheric RFs, the corresponding NRFs [W m$^{-2}$ DU$^{-1}$] are given in italics. Values are reported to 3 significant figures.**

**[1]The RF calculations for $\Delta$CC4.5(trophgt) and $\Delta$CC8.5(trophgt) employ a climate-consistent tropopause height.**

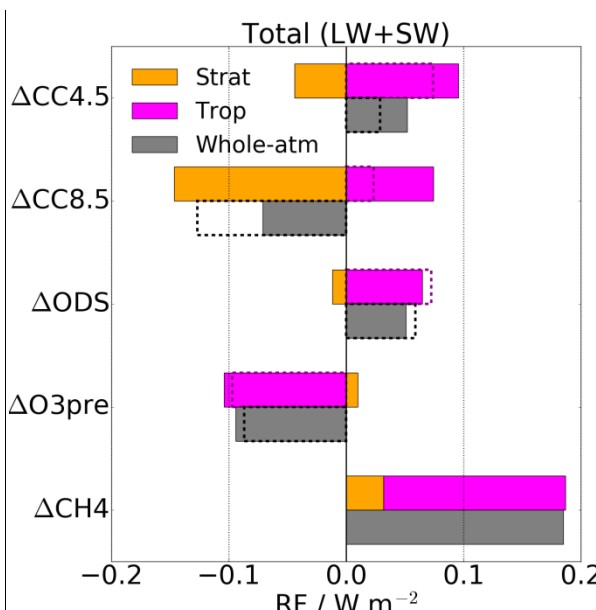

**Figure 1 – Ozone RFs [W m⁻²] due to different chemical and physical drivers for the whole-atmosphere (grey bars), stratosphere (orange bars) and troposphere (magenta bars). Dashed rectangles show RF values after tropospheric ozone changes through changes in the CH₄ lifetime are considered.**

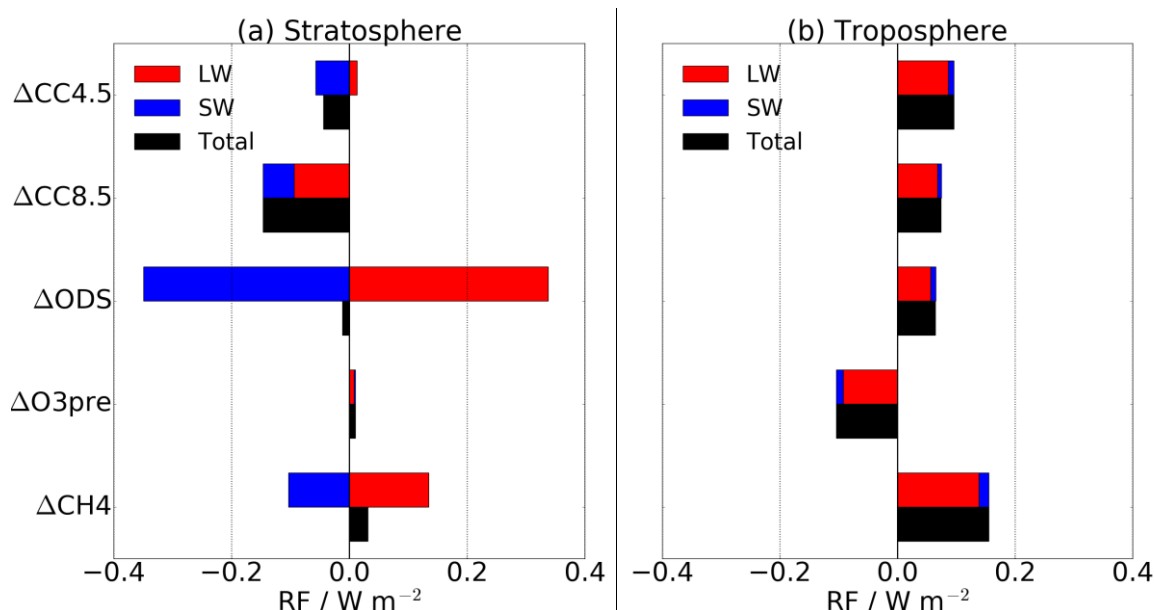

**Figure 2 - The LW (red bars), SW (blue bars) and total (LW+SW, black bars) contributions to ozone RF [W m⁻²] for changes in (a) stratospheric and (b) tropospheric ozone in each perturbation experiment. Note the change in scale from Fig. 1.**

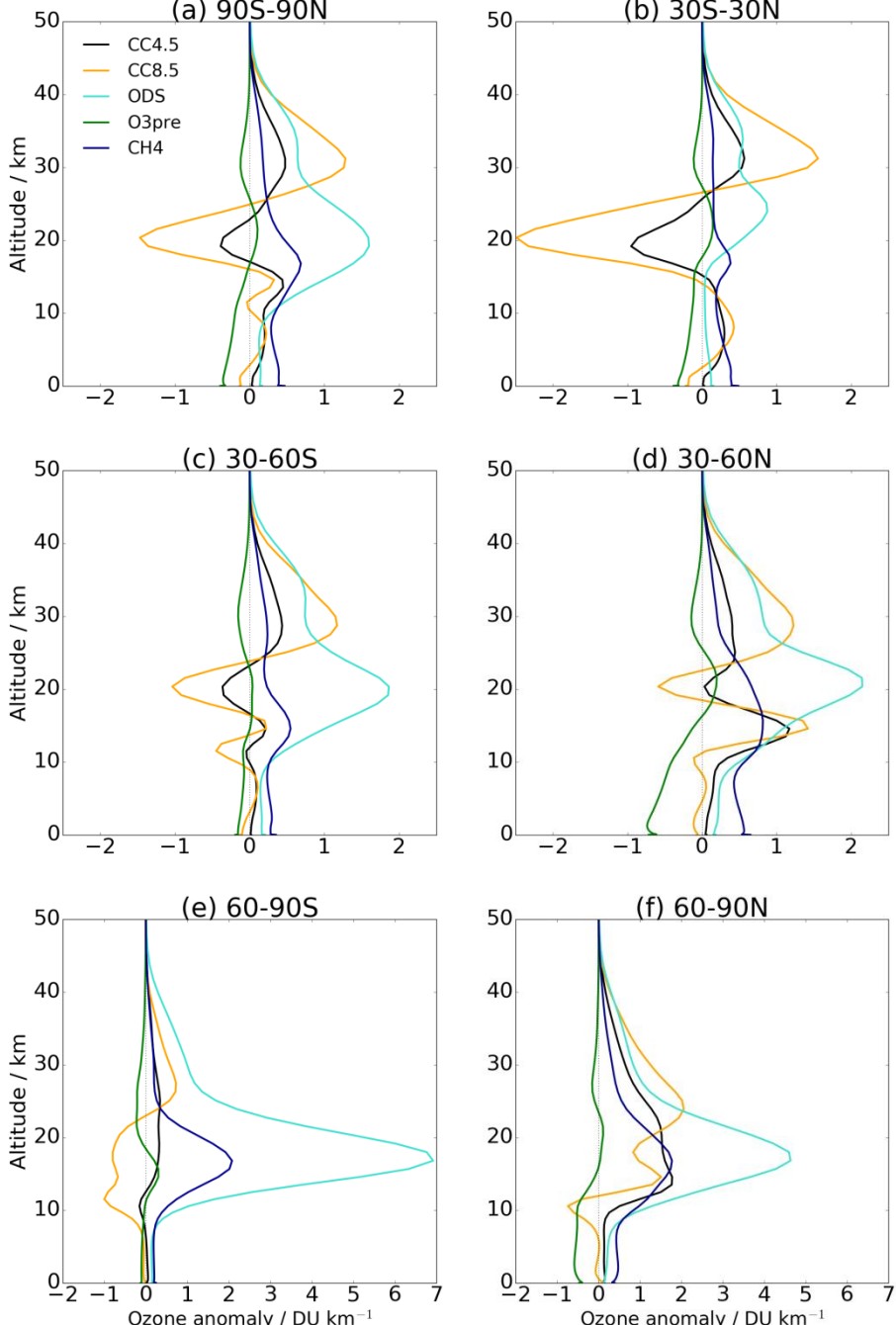

**Figure 3 – Vertical profile of annual mean ozone changes [DU km⁻¹] in each perturbation experiment relative to the Base run.** Values are averaged across 6 areas: (a) Globally (90°S-90°N), (b) Tropics (30°S-30°N), (c) Southern Hemisphere (SH) mid latitudes (30-60°S), (d) Northern Hemisphere (NH) mid latitudes (30-60°N), (e) SH high latitudes (60-90°S) and (f) NH high latitudes (60-90°N).

### 3.1 Climate change

The *sign* of the whole-atmosphere ozone RF under climate change depends on the WMGHG emissions scenario considered: a positive RF is calculated for ΔCC4.5 (+0.05 W m$^{-2}$), but a negative RF for ΔCC8.5 (-0.07 W m$^{-2}$) (Fig. 1, Table 2).

The difference between the two scenarios arises mainly from the stratospheric ozone RF, which is less negative in ΔCC4.5 (-0.04 W m$^{-2}$) than in ΔCC8.5 (-0.15 W m$^{-2}$) (Fig. 1, Table 2). Fig. 2a further shows that this difference stems from the LW, rather than the SW, contribution to RF. As Sect. 4 will discuss, the stratospheric LW contribution to RF in ΔCC8.5 is dominated by the effects of a reduction in ozone in the tropical lower stratosphere (Fig. 3b); this is driven by an increase in the upwelling mass flux by 27%, with an additional contribution from a higher tropopause also being likely. Qualitatively similar conclusions have been drawn for larger 4x$CO_2$ perturbation experiments (Dietmüller et al., 2014; Nowack et al., 2014). In contrast, ΔCC4.5 shows a small positive stratospheric LW RF (Fig. 2a). This can partly be explained by more comparable changes in tropical lower stratospheric ozone (driven by an increase in the upwelling mass flux by 10%) and upper stratospheric ozone (Fig. 3b). Indeed, in a related study focusing on tropical column ozone (Keeble et al., 2017), we find that the change in lower stratospheric ozone scales more strongly with GHG concentration than the change in upper stratospheric ozone, which is driven by cooling from $CO_2$: 0.03 versus 0.02 DU per ppmv of $CO_2$-equivalent, where $CO_2$-equivalent is the concentration of $CO_2$ that would cause the same RF as the mixture of all GHGs.

Figure 1 highlights that the RF due to tropospheric ozone changes could also be an important component of the whole-atmosphere RF due to climate change, which models without comprehensive tropospheric chemistry are unlikely to capture properly. The total tropospheric RFs are positive for both ΔCC4.5 (0.10 W m$^{-2}$, 0.040 W m$^{-2}$ DU$^{-1}$) and ΔCC8.5 (0.07 W m$^{-2}$, 0.069 W m$^{-2}$ DU$^{-1}$) and are dominated by the LW forcing (Fig. 2b; see also Rap et al. (2015)). The tropospheric ozone increase and its RF is smaller for the greater climate forcing (ΔCC8.5) due to the relatively stronger effects of tropospheric ozone reductions over ozone increases (the drivers of which are discussed below) than under a weaker climate forcing (ΔCC4.5). The tropospheric RFs outweigh (ΔCC4.5) or partly cancel (ΔCC8.5) the negative RF from stratospheric ozone changes. Consideration of $CH_4$ adjustments reduces the positive tropospheric ozone RFs by 0.02 W m$^{-2}$ (ΔCC4.5) and 0.05 W m$^{-2}$ (ΔCC8.5) (see Supplementary Material Table S1), but does not change the sign of the overall tropospheric or whole-atmosphere RFs. Note that the respective changes in $CH_4$ abundance to steady state lead to direct RFs that are larger in magnitude: -0.10 and -0.22 W m$^{-2}$ (Table S1).

A large driver of the tropospheric ozone RF is the increase in lightning $NO_x$ emissions (L$NO_x$) under climate change. We use an additional simulation that fixes L$NO_x$ to Base values within the ΔCC8.5 experimental set-up (labelled ΔCC8.5(fL$NO_x$); see Banerjee et al. (2014)) to deduce that the increase in L$NO_x$ under climate change at RCP8.5 (global total 4.7 Tg(N)yr$^{-1}$) leads to a tropospheric ozone RF of 0.24 W m$^{-2}$ (compare rows for ΔCC8.5 and ΔCC8.5(fL$NO_x$) in Table 2). The tropospheric ozone RF from L$NO_x$ is enhanced slightly by an increase in STE that is caused by a strengthened stratospheric circulation, but is offset primarily by the effects of increased humidity-driven ozone loss (Banerjee et al., 2016). The smaller tropospheric ozone RF in ΔCC8.5 compared to ΔCC4.5 is likely a result of the humidity-driven ozone

losses cancelling ozone increases in the extratropics (Fig. 3), as well as larger ozone reductions around the tropopause due to a higher tropopause (e.g. see orange line for ΔCC8.5 in Fig. 3c around 12 km).

Interestingly, the increase in $LNO_x$ is also associated with a stratospheric ozone RF of 0.04 W m$^{-2}$ (compare rows for ΔCC8.5 and ΔCC8.5(f$LNO_x$) in Table 2). This RF is consistent with increases in lower stratospheric ozone abundances following its transport from the upper troposphere (Banerjee et al., 2014). Overall, the whole-atmosphere RF is over five times larger in magnitude (-0.37 W m$^{-2}$) when $LNO_x$ is held fixed than when allowed to vary with climate change in ΔCC8.5 (-0.07 W m$^{-2}$), which points to a potentially important role of $LNO_x$ as a chemistry-climate feedback.

There is considerable inter-model spread in the tropospheric ozone response, and thus in the associated ozone RF, to climate change (Stevenson et al., 2013). The multi-model mean tropospheric ozone RF between 2000 and 2100 under RCP8.5 across 8 CCMs is a negligible value of about -0.01 W m$^{-2}$ (calculated from the final row of Table 12 in Stevenson et al. (2013) by taking the difference of the climate change-induced ozone RFs between 1850-2000 and 1850-2100). However, this reflects cancellations between larger magnitude positive and negative values for individual models: the inter-model range spans ±0.07 W m$^{-2}$. Our value of 0.07 W m$^{-2}$ is on the upper end of the inter-model range and could reflect a particularly large sensitivity of $LNO_x$ to climate in our model: 0.96 Tg(N) yr$^{-1}$ K$^{-1}$ (Banerjee et al., 2014) compared to a multi-model mean of 0.37 ± 0.06 Tg(N) yr$^{-1}$ K$^{-1}$ for the same 8 CCMs discussed above (calculated using Table S2 of Finney et al. (2016)). Our results serve to show that reducing the inter-model uncertainty in tropospheric ozone projections, and not just in stratospheric projections, is crucial for constraining the future whole-atmosphere ozone RF. Moreover, we show that the whole-atmosphere RF can result from cancellations between stratospheric and tropospheric RFs that are individually larger in magnitude. Thus, it is important to comprehensively simulate effects from both the stratosphere and troposphere to capture the climate impacts of ozone.

Finally, we note that, in order to maintain consistency with previous studies (Dietmüller et al., 2014; Nowack et al., 2014; Stevenson et al., 2013), the values of the ozone RF discussed thus far do not consider the effect of the increase in tropopause height under climate change. We calculate that employing climate consistent tropopause heights causes only small differences (≤|0.02| W m$^{-2}$) for the stratospheric, tropospheric and whole-atmosphere RFs (Table 2).

### 3.2 Reductions in ODSs

The whole-atmosphere ozone RF calculated for the ΔODS perturbation is +0.05 W m$^{-2}$ (Fig. 1, Table 2). This offsets around a quarter of the estimated direct RF of the ozone-depleting halocarbons between 2000-2100 under RCP4.5, which we estimate to be around -0.22 W m$^{-2}$ as the difference between the total halocarbon forcing (-0.15 W m$^{-2}$) (Meinshausen et al., 2011) and the non-ODS halocarbon (HFC) forcing (around +0.07 W m$^{-2}$ from Fig. 1 of Xu et al. (2013)). The future ozone RF due to ODSs is approximately a third of the estimated magnitude over the historical period (-0.15 W m$^{-2}$ between 1750-2011 (Myhre et al., 2013)), since ODS concentrations have not returned to pre-1960 values by the end of the century; note

there is a slight overlap of around a decade between our reference point (year 2000) and the historical period as defined in Myhre et al. (2013).

Despite large stratospheric ozone changes occurring in the $\Delta$ODS experiment (up to 7 DU km$^{-1}$; Fig. 3), the stratospheric ozone RF is negligible. This arises from the almost complete cancellation between two larger terms: the LW RF (mainly due to ozone increases in the lower stratosphere) and SW RF (mainly due to ozone increases in the upper stratosphere) (Fig. 2a). Note that the degree of cancellation between the LW and SW RF, and hence, the sign of the stratospheric ozone RF appears to be model dependent (Arblaster et al., 2014). This is likely due to inter-model differences in the vertical structure of the ozone response and/or in the background climatology (and hence changes in the LW component following stratospheric temperature adjustments).

The importance of the stratosphere in this experiment is found instead in the enhancment of STE by virtue of there being more stratospheric ozone available for transport; this is the primary driver of changes in tropospheric ozone in the middle and high latitudes (Fig. 1; Banerjee et al. (2016)). Consistently, we calculate a tropospheric ozone RF of +0.06 W m$^{-2}$ (Fig. 1, Table 2) or 0.035 W m$^{-2}$ DU$^{-1}$, which is enhanced by 0.01 W m$^{-2}$ when CH$_4$ adjustments are considered (alongside a direct CH$_4$ RF of 0.03 W m$^{-2}$; Table S1). We further use a "stratospheric ozone tracer" (see Banerjee et al. (2016)) to determine that ~85% of the tropospheric RF in the $\Delta$ODS experiment can be attributed to ozone of stratospheric origin, emphasizing the importance of STE for the climate effects of ozone.

### 3.3 Reductions in non-methane ozone precursor emissions

The whole-atmosphere ozone RF in $\Delta$O3pre is -0.09 W m$^{-2}$ (Fig. 1, Table 2). This arises primarily through reductions in tropospheric ozone in the northern hemisphere (see Fig. 3b, d, f) and the associated RF (-0.10 W m$^{-2}$ or 0.035 W m$^{-2}$ DU$^{-1}$). Consideration of the effects of changes in CH$_4$ abundance to steady state result in an additional indirect ozone RF of +0.01 W m$^{-2}$ and a direct CH$_4$ RF of +0.03 W m$^{-2}$ (Table S1). Nonetheless, the overall combined effect of ozone and CH$_4$ changes still represents a climate co-benefit (-0.05 W m$^{-2}$) from air pollution measures. As described previously by Banerjee et al. (2016), the changes in non-methane ozone precursor emissions do not affect stratospheric ozone abundances (see also Fig. 3). In contrast, Sect. 3.4 will show that CH$_4$ is distinct from the non-methane ozone precursors in that it *can* affect stratospheric ozone and its RF.

The ozone-derived climate effects of changes in non-methane ozone precursor emissions and CH$_4$ have often been compared (e.g. Stevenson et al., 2013; West et al., 2007). Indeed, we find in the next subsection that future increases in CH$_4$ abundance would negate the climate benefits of reductions in non-methane ozone precursor emissions. However, we here emphasise that these benefits could also be negated by future reductions in ODSs, which has previously not been noted: the whole-atmosphere ozone RF in $\Delta$ODS is over half the magnitude of the RF in $\Delta$O3pre (Fig. 1, Table 2) indicating that the combination of these perturbations would result in a smaller net ozone RF.

## 3.4 Increases in $CH_4$

The $\Delta CH4$ perturbation, in which $CH_4$ is increased from 1.75 to 3.75 ppmv following the RCP8.5 scenario, shows the largest whole-atmosphere ozone RF (0.18 W m$^{-2}$) within the set of perturbations considered (Fig. 1, Table 2). Unsurprisingly, the bulk of this RF (0.15 W m$^{-2}$, 0.036 W m$^{-2}$ DU$^{-1}$) is due to increases in tropospheric ozone, which occurs at all latitudes (Fig.

3). The ozone increase is 4.3 DU in the annual, global mean and corresponds to a sensitivity of 2.2 DU per ppmv($CH_4$), which falls within the range of other individual studies of 1.7 - 3.5 DU per ppmv($CH_4$) (Fiore et al., 2002; Kawase et al., 2011; Shindell et al., 2005; West et al., 2007).

A small fraction (~15%) of the whole atmosphere RF is due to the stratospheric ozone RF (0.03 W m$^{-2}$, Fig. 1), which is the same as the estimate in Portmann and Solomon (2007) for the same $CH_4$ increase. As for the $\Delta ODS$ experiment,

the total stratospheric RF is the result of compensating LW and SW RFs (Fig. 2a), but with a slight dominance of the LW effect over the SW in $\Delta CH4$. Correspondingly, the $\Delta CH4$ perturbation exhibits a pattern of ozone response that is similar to that for $\Delta ODS$ throughout most of the stratosphere; e.g. the perturbations to $CH_4$ (dark blue line, Fig. 3) and ODSs (light blue line, Fig. 3) both show pronounced increases in high latitude lower stratospheric ozone. The similarity arises through the common reduction in active (ozone-depleting) chlorine abundance. In $\Delta CH4$, this occurs through an increase in the

conversion of active chlorine to its reservoir, HCl, via the reaction $CH_4 + Cl \rightarrow HCl + CH_3$. There are further drivers of stratospheric ozone changes in this experiment (although we do not quantify their separate effects on ozone or the stratospheric RF): increases in lower stratospheric ozone (and hence the LW forcing) occur through $NO_x$-mediated production and transport of relatively high ozone amounts from the troposphere; increases in ozone through production of stratospheric water vapor and the consequent cooling; and reductions in ozone through greater $HO_x$-catalysed loss (Fleming

et al., 2011; Portmann and Solomon, 2007; Revell et al., 2012; Wayne, 1991). As in $\Delta ODS$, there might also be some contribution of stratospheric ozone changes to tropospheric changes through stratosphere to troposphere transport of air containing higher ozone amounts. Our estimate of the whole-atmosphere $CH_4$-driven ozone RF (0.18 W m$^{-2}$) is greater than the previous estimate of 0.13 W m$^{-2}$ in Portmann and Solomon (2007) for the same $CH_4$ increase. The difference is due to the larger tropospheric RF (0.15 versus 0.10 W m$^{-2}$); note that they did not directly diagnose the tropospheric RF due to the

simplicity of their tropospheric chemistry scheme, which could explain the difference.

There are several interactions due to time-varying emissions that are not considered in this "snapshot" experiment. Firstly, the increase in $CH_4$ is imposed under year 2000 $NO_x$ conditions. If $NO_x$ emissions were to decrease in the future, the ozone production efficiency of $CH_4$ would be reduced (Young et al., 2013), and the tropospheric ozone RF would be smaller. Secondly, the increase in $CH_4$ is imposed under year 2000 ODS loadings. As ODS loadings decrease throughout the century,

the importance of $CH_4$ in converting Cl to HCl will decrease (Fleming et al., 2011) leading to smaller stratospheric ozone changes and RF.

### 3.5 Normalised tropospheric ozone RFs

Finally, we note that the normalised ozone RF (NRF) for tropospheric ozone varies between 0.035-0.069 W m$^{-2}$ DU$^{-1}$ for the set of perturbations considered (Table 2). Low NRFs (0.035-0.045 W m$^{-2}$ DU$^{-1}$) are calculated for the ΔODS, ΔO3pre and ΔCH4 experiments. The highest values are found for the climate change scenarios: 0.045 W m$^{-2}$ DU$^{-1}$ (ΔCC4.5) and 0.069 W

m$^{-2}$ DU$^{-1}$ (ΔCC8.5). This is consistent with increases in LNO$_x$ driving ozone increases in the tropical upper troposphere where the LW radiative forcing is most sensitive to ozone changes (Rap et al., 2015). Indeed, without the increase in LNO$_x$ under climate change at RCP8.5 in the ΔCC8.5(fLNO$_x$) experiment, the NRF is only 0.045 W m$^{-2}$ DU$^{-1}$. Due to the dependence of the NRF on the vertical and latitudinal profile of the ozone change, we argue that it is inappropriate to scale the NRF (e.g. the commonly used multi-model value of 0.042 W m$^{-2}$ DU$^{-1}$ (Myhre et al., 2013)) to obtain the tropospheric

ozone RF for different emissions scenarios and different models. Instead, we demonstrate that the NRF is a useful metric to compare the efficacy of different perturbations (in a single model) to affect climate through tropospheric ozone changes; likewise, the NRF could also be used to compare the effects of the same perturbation in different models.

      The ozone RFs discussed thus far should be a good indicator of changes to the annual and global mean energy balance in response to ozone perturbations (IPCC, 2007). However, the spatially and temporally inhomogeneous nature of

these changes lead to substantial variations in RF across latitudes and seasons; these are explored in the following section.

### 4 Latitudinal and seasonal dependencies

Figure 4 shows the latitudinal distributions of the whole-atmosphere ozone RFs for the two solstice seasons: (a) June-August (JJA) and (b) December-February (DJF) for each perturbation experiment. The tropical RFs are negative for both of the climate change experiments. This can be attributed to reductions in ozone just above the tropopause (see Fig. 3b), which

result in reduced downwelling LW radiation. The negative RF in the tropics has the largest magnitude (<-0.3 W m$^{-2}$) in JJA in ΔCC8.5 (orange line, Fig. 4a); the corresponding reduction in ΔCC4.5 (black line, Fig. 4a) is ~3 times smaller.

      Interestingly, as was found for the annual and global mean RFs, even the sign of the ozone RF can depend on the WMGHG emissions scenario away from the Equator. For ΔCC4.5, positive ozone RFs in the subtropics and northern extratropics oppose the effect of ozone changes around the Equator (Fig. 4), with the net effect being a global and annual

mean positive ozone RF (Fig. 1). In contrast, the negative ozone RF in the tropics in ΔCC8.5 encompasses a wider latitude belt and is not compensated by similarly large increases elsewhere (with the exception of the subtropics in DJF; Fig. 4b), which results in a net negative global and annual mean ozone RF (Fig. 1).

      In contrast, the ΔODS experiment shows positive ozone RFs at most latitudes, contributing the largest RF in the Southern Hemisphere (SH) during JJA of the perturbations considered (light blue line, Fig. 4a) (although we note from Fig.

4b that the RF in ΔODS is reversed in sign polewards of 70°S during DJF). Further research is required to investigate the

impact of stratospheric ozone recovery, and the associated ozone RFs and climate feedbacks, on regional surface temperatures, which has been explored in only a limited number of model studies so far (Butchart et al., 2010).

In the ΔO3pre experiment (green line, Fig. 4), ozone RFs are negative across all latitudes, with a magnitude that peaks in the Northern Hemisphere (NH) subtropics and mid-latitudes in JJA. These latitudes contain the greatest reductions in precursor emissions and consequently the largest reductions in tropospheric column ozone (not shown). In JJA, the larger ozone reductions are coupled with greater temperature contrasts between the surface and upper troposphere compared to DJF (not shown), thereby enhancing the ozone RF (Haywood et al., 1998). However, all of the other perturbation experiments show positive ozone RFs in the NH extratropics, which would counteract the effect of ΔO3pre on the regional ozone RF (Fig. 4a).

Finally, the ΔCH4 experiment (dark blue line, Fig. 4) shows positive ozone RFs at almost all latitudes and in both seasons, consistent with the overall positive global mean RF (Fig. 1). As with ΔO3pre, the largest RFs are found in JJA in the NH due to greater photochemical ozone production, and an ozone increase; this likely dominates background ozone concentrations and causes a slightly larger ozone increase (and associated RF) in the SH during JJA than during DJF. Notably, by separating the chemical and radiative effects of GHGs (in particular $CH_4$), our results suggest that the future tropical ozone RF would be most influenced by the radiative effects of a large increase in GHGs, but that this would be opposed by the chemical effects of $CH_4$ (compare lines for ΔCC8.5 and ΔCH4 in Fig. 4).

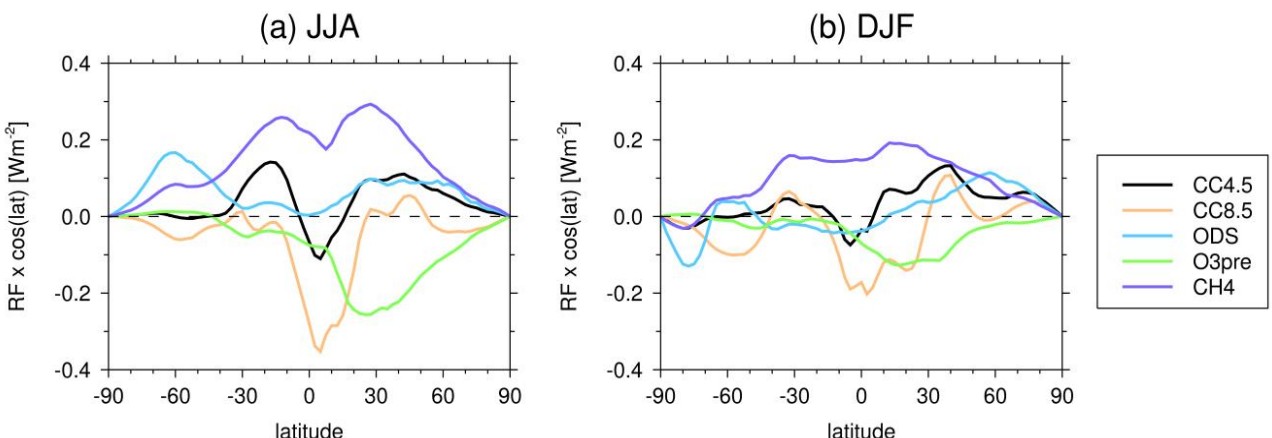

**Figure 4 - Whole-atmosphere ozone RFs [W m$^{-2}$] in (a) JJA and (b) DJF as a function of latitude for each perturbation experiment. Values have been weighted by the cosine of latitude to show the relative contributions to the global mean RFs in Fig. 1.**

**5 Conclusions**

Future changes in atmospheric ozone abundances will be determined by a complex interplay between multiple chemical and climatic drivers (e.g. Banerjee et al., 2016). This study has quantified the stratosphere-adjusted radiative forcings (RFs) associated with future changes in atmospheric ozone abundances due to different drivers using simulations from a chemistry-climate model (UM-UKCA) with a comprehensive stratospheric and tropospheric chemical scheme. We have focused on the contributions from changes in stratospheric and tropospheric ozone between year 2000 and 2100 due to changes in (i) the physical climate state (i.e. radiative effects of well-mixed greenhouse gases including SST and sea ice changes); (ii) the chemical effects of ozone depleting substances (ODSs); (iii) the chemical effects of non-methane ozone precursor emissions and (iv) the chemical effects of $CH_4$.

Projected future reductions in non-methane ozone precursor emissions result in a small global and annual mean negative ozone RF (-0.09 W m$^{-2}$) that peaks in the northern mid-latitudes during boreal summer as a result of reductions in tropospheric ozone abundances.

The climate benefits of future reductions in non-methane ozone precursors could be outweighed by the climate penalty of increases in **$CH_4$**. For the extreme case of a more than doubling in $CH_4$, as projected in the RCP8.5 emissions scenario, we find a whole-atmosphere RF of 0.18 W m$^{-2}$. Most of this RF results from tropospheric ozone increases but we also calculate some contribution of the stratospheric change (0.03 W m$^{-2}$). By separating the effects of $CH_4$ from non-methane ozone precursors, we suggest that $CH_4$ is the major driver of the historical stratospheric ozone forcing found in previous studies that considered *all* ozone precursors (Shindell et al., 2013a; Søvde et al., 2011). Note that the imposed changes in $CH_4$ are uncoupled from the radiation scheme and so do not, by design, affect atmospheric temperatures. The overall effect of an increase in $CH_4$ abundance would include a cooling of the upper stratosphere that induces an ozone increase, which we suggest might reduce the SW and total ozone RF. This component of the $CH_4$-driven ozone RF is here instead included in the $\Delta$CC8.5 simulation. We also note that the ozone response to increasing $CH_4$ will likely vary over time as the background conditions (e.g. $NO_x$ and ODS loadings) change: these impacts have not been simulated in the time-slice experiments of this study and warrant future investigation.

We find an ozone RF due to the projected decline in ODSs over the 21$^{st}$ century of +0.05 W m$^{-2}$. This RF mainly arises from increases in tropospheric ozone driven by stratosphere-to-troposphere transport of air containing higher ozone concentrations. This can be compared to the estimated RF due to ozone depletion from ODSs over the historical period of -0.15 W m$^{-2}$, of which around one third is estimated to be due to reductions in tropospheric ozone (Shindell et al., 2013a).

The RF due to ozone changes from future changes in climate state is found to be highly sensitive to the greenhouse gas (GHG) emissions scenario. In particular, we find a net positive ozone RF under RCP4.5 climate change of +0.06 W m$^{-2}$, which reflects a dominant effect from projected increases in tropospheric ozone abundances. In contrast, the estimated ozone RF is -0.07 W m$^{-2}$ under RCP8.5 climate change, which mainly reflects a larger negative RF from reductions in ozone in the tropical lower stratosphere that are driven by a strengthened Brewer-Dobson circulation. Increases in tropopause height

under climate change have a negligible ($\leq$|0.02| W m$^{-2}$) impact on ozone RFs under both the scenarios of climate change considered here.

The results emphasize that the total ozone RF over this century will result from the net effect of multiple drivers that can have distinct effects on the distributions of both stratospheric and tropospheric ozone. We recommend that future studies of ozone RF aim to attribute total (stratospheric + tropospheric) ozone RF to particular *emissions* and further separate this into *stratospheric* and *tropospheric* components, with the use of careful terminology. For example, we recommend the emissions-based view of RF in Fig. 8.17 of Myhre et al. (2013) that shows the total ozone RF for each emission ('O$_3$' bars), but with an additional quantification of 'O$_3$(strat)' and 'O$_3$(trop)' in each case. We note that the whole-atmosphere ozone RFs calculated for the perturbations considered in this study are small compared to the direct radiative effects of well-mixed GHGs between 2000-2100 for the two RCP scenarios considered: ~2 W m$^{-2}$ (RCP4.5) and ~6 W m$^{-2}$ (RCP8.5) (van Vuuren et al., 2011).

Whilst the list of drivers explored here is not exhaustive and does not include, for example, projected changes in N$_2$O, it captures many of the major factors expected to influence ozone abundances over the 21$^{st}$ century. In the presence of declining ODS levels, future changes in N$_2$O are expected to be important for determining stratospheric ozone abundances (Ravishankara et al., 2009). To our knowledge, only one study to date has investigated the indirect RF of N$_2$O through ozone (Portmann and Solomon, 2007). Using a 2D model, this study calculated a stratospheric ozone RF of 0.026 W m$^{-2}$ and a whole-atmosphere RF of 0.038 W m$^{-2}$ associated with a 150 ppbv increase in N$_2$O between 2000 and 2100. This whole-atmosphere ozone RF is smaller than found for any of the perturbations in our study. Nonetheless, the ozone response to increased N$_2$O and its associated RF could be better quantified in future studies using 3D chemistry-climate models.

**Acknowledgements**

We thank N. L. Abraham, A. T. Archibald, P. Braesicke, and P. Telford for discussions and computational support regarding UM-UKCA experiments. AB, ACM, and JAP were supported by the ERC under ACCI project no. 267760. ACM was also supported by a Postdoctoral Fellowship from the AXA Research Fund and a NERC Independent Research Fellowship (grant NE/M018199/1). This work made use of the facilities of HECToR, the UK's national high-performance computing service, which was provided by UoE HPCx Ltd at the University of Edinburgh, Cray Inc and NAG Ltd, and funded by the Office of Science and Technology through EPSRC's High End Computing Programme. This work also used the ARCHER UK National Supercomputing Service (http://www.archer.ac.uk).

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
