# Peer review of "Chemical and climatic drivers of radiative forcing due to changes in stratospheric and tropospheric ozone over the 21st century"

_Atmospheric Chemistry and Physics, 2017_

## Referee Comment (RC1) · Anonymous Referee #1 · 19 Sep 2017

I judge this to be well-written and original paper on an important issue, which represents a significant advance in understanding of the future drivers of ozone change in both the troposphere and stratosphere. I recommend acceptance after relatively minor modification.

My more important comments are indicated with a M

1:12 "tropospheric ozone precursor" – this is ambiguous, as it needs to be made clear this excludes methane (the ambiguity is emphasised by line 1:26 referring to methane as a tropospheric ozone precursor, and it also being a important result in this paper that methane is a stratospheric ozone precursor)

[Figure]

1:14, 2:5, 11:1 and elsewhere: The paper would be helped if it could be made clear when (for example) increases due to strat-trop exchange are due to there being more ozone to transport, rather than more advection doing the transport. Perhaps a terminology could be proposed that distinguishes the two?

2:7-8 It is unclear (and indeed it may be unclear in Myhre et al.) whether the forcings on line 1:30 assign all the ODS forcing to stratospheric ozone and all the ozone pre-cursor forcing to tropospheric ozone. I feel that one important result in this paper is that there may be a need for some better terminology to capture these effects.

2:20 and in addition, the role of NOx in forming nitrate aerosols (see e.g. Myhre et al)

M3:3 -3:16 I feel there needs to be more of a discussion about what is left out. It seems no aerosol forcing is included in the simulations (at least, it is not mentioned) and a more major issue that emerges later is that the authors have had to make a methodological choice – most notably the methane perturbations calculations are performed at present-day ODS concentrations, which might significantly impact the results. Although this is flagged later in the paper, I feel it is a major restriction that needs raising earlier, and returning to in the conclusions.

M4:1-2 "surface concentrations". I struggled to understand this. If, in the ODS and CH4 experiments, it is the surface concentrations that are perturbed, does this mean that the perturbation has then to propagate through the atmosphere by advection? If this is the case, given the age of air in the stratosphere is several years, a 10-year integration (line 4-13) is hardly long enough for the perturbation to impose itself (especially as the results seem to be averaged over this 10 year period). I feel sure I am misunderstanding here, and some improved clarity should help.

4:20 Stevenson et al. (2013) indicate that the ozone radiative forcing is significantly dependent on the spectral file used in the Edwards and Slingo code. Since this radiative forcing plays such an important role in this paper, it would be good practice, perhaps in the Supplementary, to be specific as to what spectral file is used here. There may be

further details of version numbers in the UM-UKCA that could be usefully documented at the same time

Table 1: Somewhere it may be good to spell out what makes up the WMGHGs (again in the Supplementary?). Some/all of the ODS are part of this? And in deltaO3pre, is the biomass burning assumed to be non-anthropogenic, as that is the implication of the label.

6-1: Since only adjusted forcings are presented (which is perfectly fine) it may be worth a note that some of the adjusted LW forcing is due to the SW-driven temperature changes – so the separation between SW and LW is not always a completely clean one.

6-7: "all" – this does not seem to be the case for dCH4 according to the table.

6-7: although not essential, adding the total column ozone change would be useful for this table.

6-14: Without going to the other paper, it is not clear what the equivalence is. Is it forcing equivalence, or stratospheric- temperature-change equivalence?

9-11: A minor point, but the "which is driven" part of this sentence might be better at the end of the sentence on line 7, where the ozone reduction is first mentioned (it would also shorten this long sentence).

10-21 This sentence implies that all halocarbons are ODS's (as otherwise what is the point of comparing them?). I might guess that a significant fraction of the 2000-2100 halocarbon forcing is from non-ODSs.

11:3 "0.03" – the table says 0.02

11:18-20 I was not sure what the logic of adding ODS and dO3Pre (but excluding CH4) was. What point was trying to be made?

11:32 (and 1: 16) A minor query about the "a third" – in the table it is (0.05/0.19) nearer

a quarter, although the third may be consistent with the fraction prior to rounding.

M12:15-17 As noted above, this is a major caveat which I think requires more flagging earlier in the paper and in the conclusions. It might help the discussion if it could be stated clearly how different the chlorine loading is between 2000 and 2100.

14:16-21 It is worth adding that this estimate of the methane effect is without the climate-change induced component of the ozone change resulting from CH4 increase (which I guess may be more like the dCC4.5 case, as methane wont strongly impact on upper stratospheric temperatures) and so the methane component could be even larger.

---

## Referee Comment (RC2) · Anonymous Referee #2 · 21 Sep 2017

General comments:

I find the paper by Banerjee et al. original, clear and very well-written, and it fits well into the scope of ACP. The paper builds on previous work in Banerjee et al. (2016), but takes it one step further by quantifying radiative forcing. Although the results are based only on a single model, the paper is original in the sense that detailed chemistry is included both for the troposphere and stratosphere, and the fact that several chemical/climatic drivers are studied. I recommend acceptance of the paper, but I also have some comments/concerns that need to be addressed first. Please see specific comments below.

[Figure]

Specific comments:

Page 1, line 15: Since RCP8.5 is considered rather extreme, it would be interesting, if possible, to have an estimate for O3 RF due to methane also for the RCP4.5 scenario. Do you expect the results from the methane perturbation experiment for RCP8.5 to be relatively linear, so that you can approximate the O3 RF due to RCP4.5 methane by scaling down the results from that experiment?

Page 2, line 29: For comparison, it would be useful to mention the forcing in 2000 from Stevenson et al.

Page 3, line 10-12: It is mentioned that there are previous studies on either tropospheric or stratospheric ozone RF. I would like to see some comparison in the Results section on how the results of those studies compare to the results obtained in this paper.

Page 4, line 12: Is 10 years spin-up enough for the ODS simulation, considering that the ODSs are only perturbed at the surface?

Page 5, line 3-5: I assume the tropopause height is higher in the climate perturbation experiments (especially in the RCP8.5). Perhaps I misunderstand something, but if the tropopause height is the same in all RF calculations, wouldn't that lead to a wrong split between tropospheric and stratospheric contribution to O3 RF?

Page 5, line 29: Figure 1 is not really discussed before page 9, after the discussion of Figs. 2 and 3. I suggest to change the order of the figures to reflect the order in which they are discussed.

Page 6, line 7: Not all cases show ozone RFs <0.1 W m-2. The methane case is ~0.2 W m-2.

Figure 3 caption: "d.p." - I assume this means "decimal points". Is that a common abbreviation?

[Figure]

Page 9, line 6-7: Could the ozone reduction in the tropical lower stratosphere be related to a higher tropopause in RCP8.5?

Page 9, line 17: On page 2, line 28 it states that Stevenson et al. got a value of -0.03 +/-0.04 W m-2 due to climate change up to 2100 under RCP8.5. Any idea why the value calculated here is so much higher (0.08 W m-2) and well outside their uncertainty range?

Page 10, line 1: Since the tropopause definition is the same in all RF calculations, wouldn't the tropospheric and stratospheric contributions be incomparable between the RCP8.5 and RCP4.5 experiments (see also earlier comment)?

Page 11, line 3: Table 2 says 0.02 and not 0.03 W m-2 DU-1.

Page 13, line 26-27: The O3 RF from the CH4 experiment is greater in JJA both in the southern and northern hemisphere. In the southern hemisphere, I would expect the photochemical ozone production to be lower during JJA than DJF?

Page 15, line 9-12: On page 6, line 8, RF values for WMGHG are 3 and 6 W m-2 for RCP4.5 and RCP8.5, respectively, and with a reference to Myhre et al. (2013). Here it is given as 2 and 6 W m-2 with a reference to van Vuuren et al. (2011). Would be good to be consistent.

Page 15, line 16-17: Is it possible to say something about how important future N2O changes may be for O3 RF, based on, if available, any estimates/indications in the literature? Would be good, if possible, to discuss the importance of this effect relative to the effects explored in the paper.

Technical corrections:

Page 1, line 12: "Wm-2" should be "W m-2". Please correct throughout the manuscript.

Figure 1 caption: Degree signs are missing from e.g., "90S-90N". Also, I cannot see that "SH" and "NH" have been defined.

Page 10, line 25: Please fix parenthesis for the reference.

---

## Author Comment (AC1) · 1 Jan 2018

We thank Reviewer 1 for their positive judgement of the manuscript and their constructive comments. We provide our responses below in blue. Line and page numbers refer to the track changed manuscript.

Please note that in the process of reviewing this manuscript, an error was corrected in the radiative forcing calculations. The implications for the results are minor: the differences for whole-atmosphere, stratospheric and tropospheric RFs are less than 0.02 W m$^{-2}$ in magnitude. The figures, tables and text (highlighted in yellow) in the revised manuscript have all been updated to reflect the corrected calculations.

I judge this to be well-written and original paper on an important issue, which represents a significant advance in understanding of the future drivers of ozone change in both the troposphere and stratosphere. I recommend acceptance after relatively minor modification. My more important comments are indicated with a M

1:12 "tropospheric ozone precursor" – this is ambiguous, as it needs to be made clear this excludes methane (the ambiguity is emphasised by line 1:26 referring to methane as a tropospheric ozone precursor, and it also being a important result in this paper that methane is a stratospheric ozone precursor)

We agree that this should be clarified. We have changed the phrase on P1L12 to 'non-methane tropospheric ozone precursor'.

1:14, 2:5, 11:1 and elsewhere: The paper would be helped if it could be made clear when (for example) increases due to strat-trop exchange are due to there being more ozone to transport, rather than more advection doing the transport. Perhaps a terminology could be proposed that distinguishes the two?

We have only mentioned stratosphere-troposphere exchange (STE) a few times and so introducing new terminology might cause confusion. Instead, we simply add a clarification in each instance of why STE is changing:

P1L14: ... which is  driven by an increase in tropospheric ozone through stratosphere-to-troposphere transport of air containing higher ozone amounts.

P13L1: ... by an increase in STE that is caused by a strengthened stratospheric circulation, ...

P14L12: The importance of the stratosphere in this experiment is found instead in the enhancment os STE by virtue of there being more stratospheric ozone available for transport;  this is the primary driver of changes in tropospheric ozone in the middle and high latitudes (Fig. 1; Banerjee et al. (2016)).

P19L18:  This RF mainly arises from increases in tropospheric ozone driven by  stratosphere-to-troposphere transport of air containing higher ozone concentrations.

2:7-8 It is unclear (and indeed it may be unclear in Myhre et al.) whether the forcings on line 1:30 assign all the ODS forcing to stratospheric ozone and all the ozone precursor forcing to tropospheric ozone. I feel that one important result in this paper is that there may be a need for some better terminology to capture these effects.

Myhre et al. (2013) do not assign all the ODS forcing to stratospheric ozone and all the ozone precursor forcing to tropospheric ozone, and they do recognize their remote effects. We have clarified on P2L4:

The emission-based estimates of historical ozone RF in Myhre et al. (2013) include the effects of changes in both stratospheric and tropospheric ozone.

We agree that careful terminology is required in all future studies. Indeed the remote effects of ODSs and ozone precursors on ozone RF are not clear in any of the figures in Myhre et al. (2013). We have inserted on P19L31:

We recommend that future studies of ozone RF aim to attribute total (stratospheric + tropospheric) ozone RF to particular *emissions* and further separate this into *stratospheric* and *tropospheric* components, with the use of careful terminology. For example, we recommend the emissions-based view of RF in Fig. 8.17 of Myhre et al. (2013) that shows the total ozone RF for each emission ('O$_3$' bars), but with an additional quantification of 'O$_3$(strat)' and 'O$_3$(trop)' in each case.

2:20 and in addition, the role of NOx in forming nitrate aerosols (see e.g. Myhre et al)

We have mentioned this briefly on P2L18:

However,  there are added complications  of further climate impacts through changes in concentrations of nitrate aerosol and  the hydroxyl (OH) radical (Myhre et al., 2013); only the latter effect is explored in this study. Changes in OH concentration  perturb ...

M3:3 -3:16 I feel there needs to be more of a discussion about what is left out. It seems no aerosol forcing is included in the simulations (at least, it is not mentioned) and a more major issue that emerges later is that the authors have had to make a methodological choice – most notably the methane perturbations calculations are performed at present-day ODS concentrations, which might significantly impact the results. Although this is flagged later in the paper, I feel it is a major restriction that needs raising earlier, and returning to in the conclusions.

We have added the following discussions:

P4L17: There are some forcings and interactions that we do not consider in this study. Firstly, our focus lies on estimating the future ozone RF from emitted gases. We do not simulate any associated aerosol forcing, with aerosol precursor emissions and their oxidant fields being held fixed in all simulations (following the scheme of Bellouin et al. (2011)). Secondly, the 'snapshot' experiments of this study do not consider various transient interactions. For example, the background conditions of NO$_x$ and ODSs affect CH$_4$ concentrations, but this coupling is not considered when perturbing NO$_x$, ODSs and CH$_4$ individually in the ΔO3pre, ΔODS and ΔCH4 experiments (potential consequences for the CH$_4$-induced ozone RF are, however, discussed in Sect. 3.4).

P19L15: We also note that the ozone response to increasing CH$_4$ will likely vary over time as the background conditions (e.g. NO$_x$ and ODS loadings) change: these impacts have not been simulated in the time-slice experiments of this study and warrant future investigation.

M4:1-2 "surface concentrations". I struggled to understand this. If, in the ODS and CH4 experiments, it is the surface concentrations that are perturbed, does this mean that the perturbation has then to propagate through the atmosphere by advection? If this is the case, given the age of air in the

stratosphere is several years, a 10-year integration (line 4-13) is hardly long enough for the perturbation to impose itself (especially as the results seem to be averaged over this 10 year period). I feel sure I am misunderstanding here, and some improved clarity should help.

Each integration is 20 years long consisting of a 10-year spin up and 10-year analysis period (P4L24). In the $\Delta$ODS and $\Delta$CH4 experiments, initial conditions of ODSs and $CH_4$, respectively, were also perturbed in order to reduce the required spin up time. Moreover, the mean age of stratospheric air is relatively short in this model (up to 4 years), so a 10-year spin up period is enough for stratospheric concentrations to reach steady-state. This was confirmed by checking the time series of long lived tracers (ODSs, $CH_4$ and $N_2O$) at various latitudes and altitudes. We have added:

P4L11: The initial atmospheric concentrations of ODSs and $CH_4$ were also perturbed by the same factor in $\Delta$ODS and $\Delta$CH4, respectively, in order to reduce spin up time.

P4L24: It was confirmed that this spin up period was long enough for stratospheric concentrations of perturbed gases to reach steady state.

4:20 Stevenson et al. (2013) indicate that the ozone radiative forcing is significantly dependent on the spectral file used in the Edwards and Slingo code. Since this radiative forcing plays such an important role in this paper, it would be good practice, perhaps in the Supplementary, to be specific as to what spectral file is used here. There may be further details of version numbers in the UM-UKCA that could be usefully documented at the same time

The names of the spectral files used in the RTM are for LW: spec3a_lw_hadgem1_wz_spec and for SW: spec3a_sw_hgem1_ln6e_mean_spec. We have added this as a footnote on Page 5.

Table 1: Somewhere it may be good to spell out what makes up the WMGHGs (again in the Supplementary?). Some/all of the ODS are part of this? And in deltaO3pre, is the biomass burning assumed to be non-anthropogenic, as that is the implication of the label.

Some (but not all) of the ODSs are radiatively active. The long-lived CFCs (CFC-11 and CFC-12) are WMGHGs and are thus included in this definition. We have added the following sentence to P4L2 and Table 1's caption: "Here, the WMGHGs considered are $CO_2$, $CH_4$, $N_2O$, CFCs, HCFCs and HFCs."

The Supplementary Material only contains Table S1, which pertains to methane feedbacks, so we do not feel a description of WMGHGs here is appropriate.

Despite biomass burning being largely of human-induced origin, it is conventionally considered as separate from anthropogenic emissions (from the combustion of fossil fuels). We follow the IPCC AR5 / ACCMIP definition in Lamarque et al. (2010):"...anthropogenic (defined here as originating from energy use in stationary and mobile sources, industrial processes, domestic and agricultural activities) and open biomass burning emissions.". We have referenced this paper in P4L5.

6-1: Since only adjusted forcings are presented (which is perfectly fine) it may be worth a note that some of the adjusted LW forcing is due to the SW-driven temperature changes – so the separation between SW and LW is not always a completely clean one.

The effect of SW-driven temperature changes is well known to be an important contribution to the adjusted LW forcing for changes in stratospheric ozone. We have mentioned this on P5L18:

The stratospheric temperature adjustment strongly affects the calculated LW (and hence total) RF for stratospheric ozone changes, with the adjustment being largest where the SW-driven temperature changes are largest (Forster and Shine, 1997).

6-7: "all" – this does not seem to be the case for dCH4 according to the table.

We thank both reviewers for pointing this out. Even considering the $\Delta CH_4$ experiment, the whole-atmosphere ozone RFs are small compared to the direct RF from WMGHGs. Hence, we have only modified the sentence on P7L1 slightly:

... the whole-atmosphere ozone RFs are small ($\ll\leq|0.12|$ W m$^{-2}$) ...

6-7: although not essential, adding the total column ozone change would be useful for this table.

We do not discuss total column ozone changes and so we would prefer to omit these values and avoid unnecessary clutter in the table.

6-14: Without going to the other paper, it is not clear what the equivalence is. Is it forcing equivalence, or stratospheric- temperature-change equivalence?

We have clarified the definition of Carbon Dioxide Equivalent; please see the amended paragraph under the next comment.

9-11: A minor point, but the "which is driven" part of this sentence might be better at the end of the sentence on line 7, where the ozone reduction is first mentioned (it would also shorten this long sentence).

We have updated the paragraph beginning P12L5 to improve coherency:

The difference between the two scenarios arises mainly from the stratospheric ozone RF, which is less negative in $\Delta CC4.5$ (-0.04 W m$^{-2}$) than in $\Delta CC8.5$ (-0.15 W m$^{-2}$) (Fig. 2Fig. 1, Table 2). Fig. 3Fig. 2a further shows that this difference stems from the LW, rather than the SW, contribution to RF. As Sect. 4 will discuss, the stratospheric LW contribution to RF in $\Delta CC8.5$ is dominated by the effects of a reduction in ozone in the tropical lower stratosphere (Fig. 1Fig. 3b); this is driven by an increase in the upwelling mass flux by 27%, with an additional contribution from a higher tropopause also being likely. Qualitatively similar conclusions have been drawn for larger $4xCO_2$ perturbation experiments (Dietmüller et al., 2014; Nowack et al., 2014). In contrast, $\Delta CC4.5$ shows a small positive stratospheric LW RF (Fig. 32a). which This can partly be explained by more comparable changes in tropical lower stratospheric ozone (driven by an increase in the upwelling mass flux by 10%) and upper stratospheric ozone (Fig. 1Fig. 3b). Indeed, in a related study focusing on tropical column ozone (Keeble et al., 2017), we find that the change in lower stratospheric ozone, which is driven by increases in the tropical upwelling mass flux (by 10 and 27% in $\Delta CC4.5$ and $\Delta CC8.5$, respectively), scales more strongly with GHG concentration (0.03 DU per ppmv of Carbon Dioxide Equivalent (CDE)) than the change in upper stratospheric ozone, which is driven by cooling from $CO_2$ (0.02 DU ppmv(CDE)$^{-1}$).: 0.03 versus 0.02 DU per ppmv of $CO_2$-equivalent, where $CO_2$-equivalent is the concentration of $CO_2$ that would cause the same RF as the mixture of all GHGs.

10-21 This sentence implies that all halocarbons are ODS's (as otherwise what is the point of comparing them?). I might guess that a significant fraction of the 2000-2100 halocarbon forcing is from non-ODSs.

This is a good point: the HFCs are greenhouse gases but are not ODSs, so we have modified the comparison (P13L31):

This offsets around  a quarter of the estimated direct RF of the ozone-depleting halocarbons between 2000-2100 under RCP4.5, which we estimate to be around -0.22 W m$^{-2}$ as the difference between the total halocarbon forcing (-0.15 W m$^{-2}$)  (Meinshausen et al., 2011) and the non-ODS halocarbon (HFC) forcing (around +0.07 W m$^{-2}$ from Fig. 1 of Xu et al. (2013)).

11:3 "0.03" – the table says 0.02

We have updated both instances with the revised and more precise values of 0.035 W m$^{-2}$.

11:18-20 I was not sure what the logic of adding ODS and dO3Pre (but excluding CH4) was. What point was trying to be made?

The effects of $\Delta$O3pre and $\Delta$CH4 have often been compared and contrasted within the literature [*West et al.*, 2007; *Stevenson et al.*, 2013]. Thus, it is recognised that the climate penalty from future increases in CH$_4$ would negate the climate benefits from reductions in non-methane ozone precursor emissions. Here, we wish to highlight the additional competing effect of ODS reductions (albeit a smaller effect than CH$_4$ increases) that has previously been overlooked. We have clarified this reasoning on P14L30:

The ozone-derived climate effects of changes in non-methane ozone precursor emissions and CH$_4$ have often been compared (e.g. Stevenson et al., 2013; West et al., 2007). Indeed, we find in the next subsection that future increases in CH$_4$ abundance would negate the climate benefits of reductions in non-methane ozone precursor emissions. However, we here emphasise that these benefits could also be negated by future reductions in ODSs, which has previously not been noted:  the whole-atmosphere ozone RF in $\Delta$O3pre $\Delta$ODS is over half the magnitude of the RF in $\Delta$O3pre  (Fig. 1, Table 2) indicating that the combination of these perturbations would result in a small net ozone RF.

11:32 (and 1: 16) A minor query about the "a third" – in the table it is (0.05/0.19) nearer a quarter, although the third may be consistent with the fraction prior to rounding.

The revised values show a smaller relative contribution of stratospheric ozone RF. We have amended the following instances:

P1L17: A  small fraction (~15%) of the ozone RF due to the projected increase in methane results from increases in stratospheric ozone.

P15L15:  A small fraction (~15%) of the whole atmosphere RF is due to the stratospheric ozone RF (0.03 W m$^{-2}$, Fig. 1), ...

M12:15-17 As noted above, this is a major caveat which I think requires more flagging earlier in the paper and in the conclusions. It might help the discussion if it could be stated clearly how different the chlorine loading is between 2000 and 2100.

This has been addressed in a previous comment. In addition, we have included the numerical changes in ODS boundary concentrations (and other species) in the caption of Table 1.

14:16-21 It is worth adding that this estimate of the methane effect is without the climate-change induced component of the ozone change resulting from CH4 increase (which I guess may be more like the dCC4.5 case, as methane wont strongly impact on upper stratospheric temperatures) and so the methane component could be even larger.

The direct impact of increased $CH_4$ on stratospheric temperatures would likely reduce the total $CH_4$-driven ozone RF: a cooling of the upper stratosphere would induce an increase in ozone and a reduction in downwelling SW radiation. We have added a qualitative statement to this effect in P19L12:

Note that the imposed changes in $CH_4$ are uncoupled from the radiation scheme and so do not, by design, affect atmospheric temperatures. The overall effect of an increase in $CH_4$ abundance would include a cooling of the upper stratosphere that induces an ozone increase, which we suggest might reduce the SW and total ozone RF. This component of the $CH_4$-driven ozone RF is here instead included in the $\Delta CC8.5$ simulation.

---

## Author Comment (AC2) · 1 Jan 2018

We are grateful to Reviewer 2 for their thoughtful comments. We provide our responses below in blue. Line and page numbers refer to the track changed manuscript.

Please note that in the process of reviewing this manuscript, an error was corrected in the radiative forcing calculations. The implications for the results are minor: the differences for whole-atmosphere, stratospheric and tropospheric RFs are less than 0.02 W m$^{-2}$ in magnitude. The figures, tables and text (highlighted in yellow) in the revised manuscript have all been updated to reflect the corrected calculations.

General comments:

I find the paper by Banerjee et al. original, clear and very well-written, and it fits well into the scope of ACP. The paper builds on previous work in Banerjee et al. (2016), but takes it one step further by quantifying radiative forcing. Although the results are based only on a single model, the paper is original in the sense that detailed chemistry is included both for the troposphere and stratosphere, and the fact that several chemical/climatic drivers are studied. I recommend acceptance of the paper, but I also have some comments/concerns that need to be addressed first. Please see specific comments below.

Specific comments:

Page 1, line 15: Since RCP8.5 is considered rather extreme, it would be interesting, if possible, to have an estimate for O3 RF due to methane also for the RCP4.5 scenario. Do you expect the results from the methane perturbation experiment for RCP8.5 to be relatively linear, so that you can approximate the O3 RF due to RCP4.5 methane by scaling down the results from that experiment?

Previous studies suggest that there is a small non-linearity in the response of tropospheric ozone to changing $CH_4$ abundance (Wild, 2007) but a fairly linear response of stratospheric ozone (Revell et al., 2012). These studies did not determine the associated linearity or lack thereof in ozone RF; for the relatively small RF values we are considering, we suspect a fairly linear relationship. However, we are unable to perform any further integrations at this stage to test this.

Page 2, line 29: For comparison, it would be useful to mention the forcing in 2000 from Stevenson et al.

We have added the forcing in 2000 (and have removed the rounding of their figures) on P2L32:

...suggests a tropospheric ozone RF of -0.033 ± 0.042 W m$^{-2}$ (multi-model mean ± 1σ) due to climate change up to 2100 under the RCP8.5 scenario, which is a negligible change from the forcing in the year 2000 of -0.024 ± 0.027 W m$^{-2}$ (both relative to 1850) (Stevenson et al., 2013).

Page 3, line 10-12: It is mentioned that there are previous studies on either tropospheric or stratospheric ozone RF. I would like to see some comparison in the Results section on how the results of those studies compare to the results obtained in this paper.

We have already compared our results to previous studies in the following instances: a qualitative similarity in the stratospheric RF between our ΔCC8.5 experiment and 4xCO2 scenarios (P12L10), a quantitative comparison of the future tropospheric RF between our climate change experiments and the multi-model results in Stevenson et al. (2013) (P13L11), the cancellation between the stratospheric SW and LW forcings in scenarios of ODS changes Arblaster et al. (2014) (P14L18). We have now added a comparison of our ΔCH4 results to Portmann and Solomon (2007):

P17L13:  A small fraction (~15%) of the whole atmosphere RF is due to the stratospheric ozone RF (0.03 W m$^{-2}$, Fig. 1), which is the same as the estimate of 0.03 W m$^{-2}$ in Portmann and Solomon (2007) for the same $CH_4$ increase.

P15L27: As in ΔODS, there might also be some contribution of stratospheric ozone changes to tropospheric changes through stratosphere to troposphere transport of air containing higher ozone amounts. Our estimate of the whole-atmosphere $CH_4$-driven ozone RF (0.18 W m$^{-2}$) is greater than the previous estimate of 0.13 W m$^{-2}$ in Portmann and Solomon (2007) for the same $CH_4$ increase. The difference is due to the larger tropospheric RF (0.15 versus 0.10 W m$^{-2}$); note that they did not directly diagnose the tropospheric RF due to the simplicity of their tropospheric chemistry scheme, which could explain the difference.

Other studies of the ozone RF have focused on the *historical* rather than the *future* RF, so it is difficult to make a like-to-like comparison. In P3L13, we have inserted the references of Portmann and Solomon (2007) (who assess drivers of future stratospheric ozone RF) and Stevenson et al. (2013) (who assess future tropospheric ozone RF) to highlight the comparisons we aim to make.

Page 4, line 12: Is 10 years spin-up enough for the ODS simulation, considering that the ODSs are only perturbed at the surface?

In the ΔODS and ΔCH4 experiments, initial conditions of ODSs and $CH_4$, respectively, were also perturbed in order to reduce the required spin up time. Moreover, the mean age of stratospheric air is relatively short in this model (up to 4 years), so a 10-year spin up period is enough for stratospheric concentrations to reach steady-state. This was confirmed by checking the time series of long lived tracers (ODSs, $CH_4$ and $N_2O$) at various latitudes and altitudes. We have added:

P4L11: The initial atmospheric concentrations of ODSs and $CH_4$ were also perturbed by the same factor in ΔODS and ΔCH4, respectively, in order to reduce spin up time.

P4L24: It was confirmed that this spin up period was long enough for stratospheric concentrations of perturbed gases to reach steady state.

Page 5, line 3-5: I assume the tropopause height is higher in the climate perturbation experiments (especially in the RCP8.5). Perhaps I misunderstand something, but if the tropopause height is the same in all RF calculations, wouldn't that lead to a wrong split between tropospheric and stratospheric contribution to O3 RF?

There are advantages and disadvantages of employing a fixed tropopause height in the RF calculations. The advantage is that it facilitates a like-to-like comparison with previous studies that have made the same choice (Dietmüller et al., 2014; Nowack et al., 2014; Stevenson et al., 2013). A fixed tropopause height also maintains the same mass of air in the troposphere and stratosphere so that attribution of the ozone RF (to its changing concentration/distribution) is not confounded by changing air mass. However, a fixed tropopause does not consider the changing split between stratospheric and tropospheric ozone, as the reviewer points out. We have investigated impacts of a rising tropopause under climate change, and find only small effects, which we highlight in the following instances:

Table 2: added two rows (ΔCC4.5(trophgt) and ΔCC8.5(trophgt)).

P1L22: Considering the increases in tropopause height under climate change causes only small differences (≤|0.02| W m$^{-2}$) for the stratospheric, tropospheric and whole-atmosphere RFs.

P5L22: In the climate change experiments, ΔCC4.5 and ΔCC8.5, the tropopause rises; the ramifications for employing a climate-consistent tropopause height for the ozone RF will be shown to be small (see Sect. 3.1).

P14L30: Finally, we note that, in order to maintain consistency with previous studies (Dietmüller et al., 2014; Nowack et al., 2014; Stevenson et al., 2013), the values of the ozone RF discussed thus far do not consider the effect of the increase in tropopause height under climate change. We calculate that employing climate consistent tropopause heights causes only small differences ($\leq |0.02|$ W m$^{-2}$) for the stratospheric, tropospheric and whole-atmosphere RFs (Table 2).

P19L27: Increases in tropopause height under climate change have a negligible ($\leq |0.2|$ W m$^{-2}$) impact on ozone RFs under both the scenarios of climate change considered here.

Page 5, line 29: Figure 1 is not really discussed before page 9, after the discussion of Figs. 2 and 3. I suggest to change the order of the figures to reflect the order in which they are discussed.

We have changed the order of the figures such that Figure 1 shows the total ozone RFs, Figure 2 shows the LW and SW RF components, then Figure 3 shows the vertical ozone profiles.

Page 6, line 7: Not all cases show ozone RFs <0.1 W m-2. The methane case is ~0.2 W m-2.

We thank both reviewers for pointing this out. Even considering the ΔCH4 experiment, the whole-atmosphere ozone RFs are small compared to the direct RF from WMGHGs. Hence, we have only modified the sentence on P7L1 slightly:

... the whole-atmosphere ozone RFs are small ($<\leq|0.\mathbf{1}2|$ W m$^{-2}$) ...

Figure 3 caption: "d.p." - I assume this means "decimal points". Is that a common abbreviation?

We think the figure is clearer without the rounding so have updated the figure and removed the abbreviation.

Page 9, line 6-7: Could the ozone reduction in the tropical lower stratosphere be related to a higher tropopause in RCP8.5?

Yes, a part of this ozone reduction will be related to a higher tropopause, though the impact is difficult to separate from the effects of strengthening tropical lower stratospheric upwelling. We have included a qualitative note (P12L8):

; this is driven by an increase in the upwelling mass flux by 27%, with an additional contribution from a higher tropopause also being likely.

Page 9, line 17: On page 2, line 28 it states that Stevenson et al. got a value of -0.03 +/-0.04 W m-2 due to climate change up to 2100 under RCP8.5. Any idea why the value calculated here is so much higher (0.08 W m-2) and well outside their uncertainty range?

-0.03 W m$^{-2}$ is the ozone RF due to climate change between 1850-2100. We discuss in P13L12 that the RF between 2000-2100 (RCP8.5) can be calculated from Table 2 in Stevenson et al. (2013) as ~0.01 W m$^{-2}$ with an inter-model range of ±0.07 W m$^{-2}$. Our calculated result of 0.07 W m$^{-2}$ lies on the upper bound of this inter-model range, and could be due to a larger sensitivity of LNO$_x$ to surface temperature in our model. We have inserted (P13L16):

Our value of 0.07 W m$^{-2}$ is on the upper end of the inter-model range and could reflect a particularly large sensitivity of LNO$_x$ to climate in our model: 0.96 Tg(N) yr$^{-1}$ K$^{-1}$ (Banerjee et al., 2014) compared to a multi-model mean of 0.37 ± 0.06 Tg(N) yr$^{-1}$ K$^{-1}$ for the same 8 CCMs discussed above (calculated using Table S2 of Finney et al. (2016)).

Page 10, line 1: Since the tropopause definition is the same in all RF calculations, wouldn't the tropospheric and stratospheric contributions be incomparable between the RCP8.5 and RCP4.5 experiments (see also earlier comment)?

Please see response to earlier comment.

Page 11, line 3: Table 2 says 0.02 and not 0.03 W m-2 DU-1.

We have updated both instances with the revised and more precise values of 0.035 W m$^{-2}$.

Page 13, line 26-27: The O3 RF from the CH4 experiment is greater in JJA both in the southern and northern hemisphere. In the southern hemisphere, I would expect the photochemical ozone production to be lower during JJA than DJF?

On increasing methane, the pattern of tropospheric column ozone increase in the SH resembles the climatological seasonal cycle. The higher column ozone (and its increases in the ΔCH4 experiment) just south of the Equator in JJA is likely due to greater interhemispheric transport from the NH (since the bulk of ozone production occurs in JJA in the NH). We have added this suggestion (P17L17):

As with ΔO3pre, the largest RFs are found in JJA in the NH due to greater photochemical ozone production, and an ozone increase, during this season; this likely dominates background ozone concentrations and causes a slightly larger ozone increase (and associated RF) in the SH during JJA than during DJF.

Page 15, line 9-12: On page 6, line 8, RF values for WMGHG are 3 and 6 W m-2 for RCP4.5 and RCP8.5, respectively, and with a reference to Myhre et al. (2013). Here it is given as 2 and 6 W m-2 with a reference to van Vuuren et al. (2011). Would be good to be consistent.

We thank the reviewer for pointing this out. The correct values are 2 and 6 W m-2 as shown by Fig. 10 in van Vuuren et al. (2011). We have amended P7L2:

(roughly 3 2 and 6 W m$^{-2}$ for RCP4.5 and RCP8.5, respectively, as shown by Fig. 10 in (Myhre et al., 2013) van Vuuren et al. (2011)).

Page 15, line 16-17: Is it possible to say something about how important future N2O changes may be for O3 RF, based on, if available, any estimates/indications in the literature? Would be good, if possible, to discuss the importance of this effect relative to the effects explored in the paper.

The final line of the manuscript mentions Portmann and Solomon (2007), which, to our knowledge, is the only study that has calculated the indirect RF of N$_2$O through ozone. We have expanded this discussion (P21L16):

The contribution of this effect to future ozone RF over the 21$^{st}$ century may also be important To our knowledge, only one study to date has investigated the indirect RF of N$_2$O through ozone (Portmann and Solomon, 2007). Using a 2D model, this study calculated a stratospheric ozone RF of 0.026 W m$^{-2}$ and a whole-atmosphere RF of 0.038 W m$^{-2}$ associated with a 150 ppbv increase in N$_2$O between 2000 and 2100. This whole-atmosphere ozone RF is smaller than found for any of the perturbations in

our study. Nonetheless, the ozone response to increased N$_2$O and its associated RF could be better quantified in future studies using 3D chemistry-climate models.

Technical corrections:

Page 1, line 12: "Wm-2" should be "W m-2". Please correct throughout the manuscript.

Corrected.

Figure 1 caption: Degree signs are missing from e.g., "90S-90N". Also, I cannot see that "SH" and "NH" have been defined.

Corrected.

Page 10, line 25: Please fix parenthesis for the reference.

Corrected.